# Epidemiology of Splenic Lesions in Dogs Undergoing Splenectomy—Pathological Characterization and Risk Factors

**DOI:** 10.3390/vetsci13010064

**Published:** 2026-01-09

**Authors:** Filippo Dell’Anno, Lucia Minelli, Giuseppe Giglia, Elvio Lepri, Marta Mechelli, Livia De Paolis, Floriana Fruscione, Elisabetta Razzuoli, Elisabetta Manuali

**Affiliations:** 1 National Reference Center of Veterinary and Comparative Oncology (CEROVEC), Istituto Zooprofilattico Sperimentale del Piemonte, Liguria e Valle d’Aosta, Piazza Borgo Pila 39/24, 16129 Genoa, Italy; filippo.dellanno@izsplv.it (F.D.); livia.depaolis@izsplv.it (L.D.P.); floriana.fruscione@izsplv.it (F.F.); 2Department of Veterinary Medicine, University of Perugia, Via S. Costanzo 4, 06126 Perugia, Italy; elvio.lepri@unipg.it; 3Istituto Zooprofilattico Sperimentale dell’Umbria e delle Marche “Togo Rosati”, Via G. Salvemini 1, 06126 Perugia, Italy; l.minelli@izsum.it (L.M.); m.mechelli@izsum.it (M.M.); e.manuali@izsum.it (E.M.)

**Keywords:** canine spleen, hemangiosarcoma, survival analysis, odds ratio

## Abstract

Splenic diseases are common in dogs and may be either non-neoplastic or neoplastic. In this retrospective study, we examined 682 canine spleens collected between 2014 and 2023. More than half of the lesions were non-neoplastic, while hemangiosarcoma was the most frequent tumor, representing over half of all neoplasms. Large breeds and some purebreds, such as Lagotto Romagnolo, Boxer, Labrador and German Shepherd, showed a significantly higher risk of developing hemangiosarcoma. Dogs with hemangiosarcoma had much shorter survival times compared with those with non-neoplastic lesions. These findings improve our understanding of the epidemiology and prognosis of splenic diseases in the canine population.

## 1. Introduction

Splenic mass lesions are often observed in dogs and represent a clinically relevant condition due to their potential to rupture, causing severe, often fatal, internal bleeding. Additionally, malignant tumors can metastasize via the blood or intra-coelomic route following rupture, reproducing the tumor in secondary sites, including vital organs (i.e., heart, liver, lung, and brain). In recent years, the availability and use of advanced imaging modalities, such as ultrasound and computed tomography, have increased the detection rate of both incidental and symptomatic splenic masses [1].

Historically, splenic masses in dogs have been associated with a high likelihood of malignancy, leading to the widely cited “two-thirds rule”, according to which most splenic masses were presumed malignant and predominantly hemangiosarcomas (HSAs). While this assumption remains partially applicable in specific clinical scenarios such as non-traumatic hemoperitoneum [2], a growing body of evidence suggests that non-neoplastic lesions may represent the majority of splenic masses identified in dogs [3,4].

Despite this growing body of literature, reported proportions of benign versus malignant splenic lesions vary widely among studies, largely reflecting differences in study design, referral patterns, and sample size. As a result, several assumptions commonly applied in clinical decision-making remain supported by heterogeneous and sometimes underpowered datasets.

Among non-neoplastic splenic lesions in dogs, the most reported include nodular hyperplasia (i.e., lymphoid, complex nodular hyperplasia), hematoma, infarction, splenitis, atrophy, and splenic congestion [5,6]. Although generally benign, some of these conditions have overlapping clinical and imaging features as well as macroscopic appearance, requiring splenectomy and subsequent histopathology for definitive diagnosis [7].

Neoplastic lesions, both benign and malignant, can arise from various cellular lines, including, but not limited to, vascular (e.g., hemangioma, HSA), lymphoid (e.g., lymphoma), fibrous (e.g., fibrosarcoma), smooth muscle (e.g., leiomyoma/leiomyosarcoma); additionally, some neoplasms can develop with an unusual cellular differentiation, including myeloid, adipose (e.g., lipoma), and bone (e.g., osteosarcoma) differentiation. Different studies have consistently identified and confirmed HSA as the most common malignant neoplasm of the canine spleen, accounting for approximately 50–74% of all splenic malignancies [3,8,9,10,11,12,13,14]. This tumor primarily affects older dogs and is associated with a highly aggressive clinical course characterized by rapid growth, frequent ruptures leading to hemoperitoneum and hypovolemic shock, and high metastatic potential with predilection for liver, lung, omentum, mesentery, and brain [15,16]. While the pathological spectrum of splenic lesions is well described, their relative frequencies and associated demographic risk factors remain inconsistently reported across retrospective studies, particularly those based on limited case numbers or single-center experiences.

Given the diagnostic challenges and prognostic implications, the histopathologic examination remains the gold standard for the classification of splenic lesions. It allows detailed assessment of tissue architecture, identification of cell atypia, mitotic count, vascular invasion, and other features critical for differentiating benign from malignant processes [16]. Retrospective studies have provided valuable insights into the epidemiology and histopathological spectrum of splenic tumors in dogs, suggesting that certain demographic and environmental variables, such as breed, age, sex, size, and living environment, may influence the occurrence and biological behavior of these lesions [13,17]. Despite the extensive literature on canine splenic lesions, including several retrospective studies and one large historical cohort [11], important knowledge gaps remain. Most available data are derived from heterogeneous populations, limited sample sizes, or outdated cohorts that do not fully reflect current diagnostic standards, population structure, and clinical management in veterinary oncology. Moreover, few studies have simultaneously integrated standardized histopathological diagnoses with detailed demographic variables and survival outcomes in a contemporary canine population.

Given the substantial evolution of diagnostic criteria, clinical awareness, and referral patterns over the past decades, an updated large-scale epidemiological assessment is warranted. Therefore, the aim of the present study was to investigate the frequency, pathological spectrum, and epidemiological distribution of splenic lesions in dogs undergoing splenectomy over a ten-year period (2014–2023), using a large, regionally representative, histopathology-confirmed cohort. In addition, we sought to evaluate demographic and environmental risk factors associated with specific lesion types and to assess their impact on clinical outcome. By providing updated effect estimates and a comprehensive comparison with previous studies, this work aims to refine current assumptions and support evidence-based clinical decision-making.

## 2. Materials and Methods

### 2.1. Data Collection

A retrospective study was carried out on histopathologic diagnoses of surgically resected canine spleens and included a total of 682 dogs diagnosed with splenic lesions (non-neoplastic and neoplastic) selected from the archive of the Canine Cancer Registry (CCR) of the Umbria Region, Italy [18]. The study analyzes a dataset on the frequency and risk factors for different types of splenic lesions in the canine population sampled in the Umbria region between 2014 and 2023. Informed consent for the use of clinical data and samples for research was obtained from the owners of dogs. In addition, clinical staging data were also retrieved following the work proposed by De Nardi and colleagues [19]. Patient records were reviewed for breed, sex, age at diagnosis, size, habitat, and overall survival. The dog population was stratified by age in five sub-categories: 0–5 years (group 1), 6–10 years (group 2), 11–15 years (group 3), 16–20+ (group 4). For the breed size, categories (small, medium, and large) were grouped according to Ente Nazionale della Cinofilia Italiana (ENCI) classification (Available from: https://www.fci.be/en/Nomenclature (accessed on 1 April 2024)). The data were described using absolute frequencies and percentages. This subdivision allowed for subsequent analysis of interactions between the size factor of the dogs and the splenic diagnosis. Follow-up data were obtained from medical records or by telephone conversation with the owners. Follow-up date was available for every dog included in the study. Survival time was defined as the time intercurrent from the histologic diagnosis date to the follow-up collection date or time to death.

Tissue samples were, at the time of submission, fixed in 10% neutral-buffered formalin and routinely embedded in paraffin wax. The 4 µm thin sections were stained with Haematoxylin and Eosin (HE) and observed with the microscope Eclipse Ci-L (Nikon Corporation, Tokyo, Japan). The HE-stained sections were evaluated in blind conditions by two to three pathologists (GG, EL, EM) for diagnostic accuracy. Each diagnosis was issued and sent to the referring veterinarian in a pathology report and automatically registered in the CCR only when two pathologists reached full agreement. In cases of disagreement, a third pathologist blindly re-reviewed the slides and then discussed the cases with the first two pathologists examining the slides until a final agreement was reached, as required by the protocol of the CCR of Umbria Region [18]. This process ensures a high level of agreement and a reliable final diagnosis. Diagnosis was based on the WHO Histological classification of hematopoietic tumors of domestic animals, second edition [20]. HSAs were diagnosed as follows: in brief, four histological parameters for neoplastic endothelial cells were assessed: cellular differentiation, nuclear pleomorphism, mitotic count, and percentage of necrosis [21]. Cellular differentiation was defined based on the ability of neoplastic endothelial cells to form well-recognizable vascular-like structures. Nuclear pleomorphism, a measure of cellular atypia, was defined based on the uniformity of nuclear size and shape between tumor cells. Finally, the mitotic count was calculated by counting all mitotic figures in 2.37 mm^2^. In case of poorly differentiated tumors, an immunohistochemical panel was applied according to needs, including one or more of the following antibodies: anti-Von Willebrand Factor (FVIII—Polyclonal rabbit anti-human Von Willebrand Factor, code number GA527, DAKO, Santa Clara, CA, USA), anti-CD31 (Monoclonal Mouse Anti-Human CD31, clone JC/70AM0823, ABCAM Limited, Cambridge, UK) and anti-Vascular Endothelial Growth Factor (VEGF—cat. number M7273, DAKO, Santa Clara, CA, USA). For the present study, all data were collected retrospectively from the existing CCR archive. No histochemical or immunohistochemical techniques were newly applied for the purposes of this study, as all diagnoses had already been established as part of the routine diagnostic workflow prior to data extraction.

### 2.2. Statistical Analysis

Descriptive analysis was conducted on the entire dataset of dogs included in the study. Descriptive analysis was performed to determine the distribution of different lesions based on demographic and environmental factors.

For each dog, the date of diagnosis, the date of the last follow-up or death, and the follow-up status were recorded. Survival analysis was performed using the Kaplan–Meier method, estimating survival curves and comparing them between groups using the log-rank test. Subsequently, a multivariate analysis was conducted using the Cox regression model, calculating hazard ratios (HR) with 95% confidence intervals. The proportional hazards assumption was tested using the Schoenfeld test. If the proportionality assumption was violated, a Cox model was applied to account for time-dependent effects. HSA was considered, for the analysis, as the major comparative group based on the worst biological behavior reported in the literature for this lesion.

Finally, the analysis of the probability of developing tumors in relation to risk factors was performed using both univariate and multivariate logistic regression analyses. Logistic regression was selected because the outcomes of interest—presence of hemangiosarcoma (HSA) vs. all other lesions, and presence of non-tumoral (NT) lesions vs. all other lesions—were binary categorical variables. The independent variables included age class, sex, breed type (purebred vs. mixed breed), habitat (urban, rural, other), and body size category (small, medium, large). Age and size groups were included as categorical variables using the lowest class as reference. Results were expressed as odds ratios (OR) with 95% confidence intervals. All analyses were conducted using STATA 17.0 (StataCorp, College Station, TX, USA).

## 3. Results

### 3.1. Descriptive Analysis

The descriptive analysis performed to provide an epidemiological overview of the population revealed that for the 682 cases analyzed, male dogs accounted for 357 cases (52.3%), while 325 cases (47.7%) were female. Out of the total, 420 cases (61.6%) occurred in purebred while 262 cases (38.4%) occurred in mixed breed dogs, according to the Ente Nazionale Cinofilia Italiana (ENCI) classification (https://www.enci.it/libro-genealogico/razze (accessed on 1 April 2024)). ENCI classification by size showed that 44.4% of dogs belonged to the medium size group (303), followed by large size with a 34.2% (233) of dogs, and small size with 21.4% (146) of dogs. The breed that accounted for the highest number were German Shepherd (54 cases,12.8%), Labrador Retriever (47 cases, 11.2%), English setter (23 cases, 5.5%), Rottweiler (20 cases, 4.8%), Golden Retriever, –English Cocker Spaniel, –Beagle (16 cases/breed, 3.8%), Jack Russel Terrier (14 cases, 3.3%) Boxer (13 cases, 3.1%) and English Springer Spaniel–Pinscher (12 cases/breed, 2.8%). Regarding sex, 325 dogs (47.6%) were females and 357 dogs (52.4%) were males. The distribution of the canine population based on the area of residence showed that most dogs lived in urban areas (440; 64.5%), followed by rural areas (159; 23.3%) and others (83; 12.2%).

Furthermore, the age at diagnosis in the population examined, and the overall mean age at diagnosis for splenic lesions was 10.4 years (SD 2.9 years). The oldest patient was 21 years old at the time of splenectomy with a diagnosis of myelolipoma, while the youngest patient was 1 month old, and the histological diagnosis revealed the presence of nodular hyperplasia. Similarly, in dogs with non-neoplastic splenic lesions, the mean age at diagnosis was 10.4 (SD 3.11 years). In dogs with HSA, the mean age at diagnosis was slightly higher, with a mean of 10.6 years, a standard deviation of 2.10 years (median: 11 years, IQR of 3 years), and the age ranged from 4 to 15 years. Descriptive analysis revealed that more than half of the dogs (370/682, 54.3%) had non-neoplastic lesions. A summary of the descriptive data related to splenic lesions, both non-neoplastic and neoplastic, can be found in Table 1.

Descriptive analysis, performed to provide a detailed overview of splenic lesions, revealed that for non-neoplastic lesions, the most frequent diagnoses were hematomas and complex follicular hyperplasia. Among the neoplastic lesions observed, HSA was the most common lesion, diagnosed in 170 dogs (54.5%), followed by other sarcomas (including histiocytic sarcomas, spindle cell sarcomas, and round cell sarcomas) in 59 dogs (19%) and lymphomas in 47 dogs (15.1%). For HSA, 11 dogs showed metastases: 2 had metastases to the lung and liver, 5 to the omentum, one to the liver and adrenal gland, 2 to the liver alone, and one involving the lung, kidney, and intestine. A single case of splenic sarcoma was detected as metastatic in the urinary bladder. Finally, four carcinomas metastasizing to the spleen were found, although the primary site was unknown. Details on the histopathological diagnoses are shown in Table 2.

The percentage distribution of the most common histological diagnoses is further reported in graphs divided by sex (Figure 1A), habitat (Figure 1B), age group (Figure 1C), and size (Figure 1D). The frequencies of the non-neoplastic lesions group (370) and the HSA (170), as the single most common neoplastic lesion, were evaluated in relation to sex, habitat, age category, and breed size. Regarding non-neoplastic lesions, males were slightly more affected than females, with 208 male (56.2%) and 162 female (43.8%) dogs. By habitat, most dogs with non-neoplastic lesions lived in urban environments (236, 63.8%), followed by rural areas (89, 24.1%) and other habitats (45, 12.2%). Analysis by age group showed that non-neoplastic lesions were most frequent in group 3 (183 dogs, 49.4%), followed by group 2 (141 dogs, 38.1%), group 1 (29 dogs, 7.8%), and group 4 (17 dogs, 4.6%). Finally, considering the breed size, medium breeds had the highest proportion of non-neoplastic lesions (173, 46.8%), followed by large breeds (114, 30.8%) and small breeds (83, 22.4%). Regarding HSA, 82 dogs were females (50.9%) and 79 males (49.1%). By habitat, the majority of HSAs occurred in urban areas (72.0%), followed by rural areas (22.4%). Analysis by age group showed for the HSA the highest proportion in group 3 (46.0%), followed by group 5 (29.8%), group 2 (17.4%), and group 4 (12.4%). Finally, examining breed size, large breeds were most frequently affected (46.6%) by HSA, followed by medium-sized breeds (37.3%) and small breeds (16.1%).

To investigate to a greater extent the canine splenic HSA, a major concern when considering canine splenic mass lesions, a deeper descriptive analysis was performed. Analyzing the distribution of the HSA by specific breed: mixed-breed dogs represented the largest group with 51 dogs (31.7%), followed by German Shepherds (21, 13.0%) and Labrador Retrievers (19, 11.8%). Breeds such as Lagotto Romagnolo, Beagle, and Golden Retriever each accounted for between 3.1% and 3.7%. Considering the distribution by size, HSA was more common in large dogs, with 75 cases (46.6%), followed by medium-sized dogs (60, 37.3%) and small dogs (26, 16.1%). The distribution of HSA between the sexes showed 82 females (50.9%) and 79 males (49.1%).

To explore the magnitude of association between demographic factors and HSA occurrence, relative risks (RRs) of HSA development were calculated for breed, size and sex. Among the breeds analyzed, the Lagotto Romagnolo had the highest risk, with an RR of 3.32 (95% CI: 1.98–5.56), indicating that these dogs were more than three times as likely to develop HSA than mixed breeds. Other groups with RRs greater than two included the Boxer (2.09; 95% CI: 0.99–4.43), Labrador Retriever (2.07; 95% CI: 1.36–3.16), and German Shepherd (2.00; 95% CI: 1.32–3.01). Breeds such as the Cane Corso (1.99; 95% CI: 0.84–4.67), Border Collie (1.79; 95% CI: 0.74–4.31), Beagle (1.65; 95% CI: 0.80–3.43), and Golden Retriever (1.65; 95% CI: 0.80–3.43) showed an increased but less marked risk. Other breeds, including Yorkshire Terriers (1.60; 95% CI: 0.55–4.59), English Cocker Spaniels (1.44; 95% CI: 0.63–3.26), and English Springer Spaniels (1.37; 95% CI: 0.54–3.48) showed a moderate increase in risk, while breeds such as Bernese Mountain Dogs and English Pointers had an RR of 1.28, and Dachshunds showed a risk almost equivalent to mixed breeds (1.09; 95% CI: 0.26–4.62). Conversely, some breeds showed a relative risk lower than 1, suggesting a lower predisposition to HSA than mixed breeds. Among these, the Rottweiler had an RR of 0.85 (95% CI: 0.32–2.28), while Pinschers (0.59; 95% CI: 0.13–2.71), English Setters (0.56; 95% CI: 0.17–1.83), and Jack Russell Terriers (0.51; 95% CI: 0.11–2.38) showed a reduced risk. Breeds such as the Maremma Sheepdog and Pitbull had an RR of 0.43 (95% CI: 0.03–6.12), while American Staffordshire Terriers and Cavalier King Charles Spaniels had an RR of 0.36 (95% CI: 0.02–5.33). Finally, the Maltese showed the lowest risk among the breeds analyzed, with an RR of 0.28 (95% CI: 0.02–4.24). Data regarding the RRs of HSA development per breed is reported in Table 3 and Figure 2.

HSA relative risk analysis based on size showed that large breed dogs had the highest risk of developing it, using medium-sized dogs as the reference group (RR = 1.0). Large breed dogs had a relative risk (RR) of 1.63 compared to medium-sized dogs. Small dogs showed a slightly lower risk corresponding to an RR of 0.90. HSA relative risk analysis based on sex showed that the risk was slightly higher in females than in males. The relative risk calculated using females as the reference group was 1 for females and 0.91 for males, indicating a slight reduction in risk in males.

### 3.2. Survival Analysis

To evaluate survival patterns in dogs diagnosed with different splenic lesions, a survival analysis was conducted, showing significant differences. Kaplan–Meier analysis revealed markedly reduced survival in dogs with HSA compared to non-tumor (NT) lesions (log-rank χ^2^ (1) = 32.98, *p* < 0.0001) (Figure 3). Observed events were 164 in HSA versus 366 in NT lesions. Univariate Cox regression, used to assess the effect of demographic variables on survival, confirmed the survival differences. HSA was associated with a significantly increased risk of event occurrence compared to NT lesions (HR = 1.93, 95% CI: 1.53–2.43, *p* < 0.0001; LR χ^2^ (1) = 28.35, *p* < 0.0001). Proportional hazards assumptions were violated for HSA (Schoenfeld residuals, global χ^2^ (1) = 20.50, *p* = 0.0000), indicating that the effects of lesion type on survival were time dependent. Time-varying Cox models incorporating follow-up time confirmed these dynamics. HSA carried a higher early hazard (HR = 3.42, 95% CI: 2.47–4.75, *p* < 0.0001) with decreasing risk over time (tvc HR < 1, *p* < 0.001). Stratified interval analyses further supported a non-proportional, time-dependent effect of lesion type on survival. No significant difference in survival time was observed between the other neoplastic lesions and the non-tumoral lesion.

Kaplan–Meier survival curves were generated to compare outcomes among dogs diagnosed with different tumor types (Figure 4). After excluding non-tumor lesions (NT), dogs diagnosed with malignant neoplastic lesions (hemangiosarcoma (HSA), lymphoma and other sarcomas) were included in the survival analyses.

Comparison of survival curves using the log-rank test revealed significant differences across tumor categories (χ^2^ = 10.87, *p* = 0.0044). Dogs with hemangiosarcoma exhibited the poorest survival probability, with curves declining more steeply than all other tumor groups.

To further quantify differences in mortality risk between tumor types, a Cox proportional hazards model was fitted using hemangiosarcoma as the reference category. The model was significant (LR χ^2^ = 11.87, *p* = 0.0026), highlighting, for lymphoma, a significantly lower risk of death compared with HSA. Specifically, lymphoma was associated with a 52% reduction in hazard (HR = 0.48, 95% CI: 0.31–0.75, *p* = 0.001). Category “Other Sarcomas” exhibited a non-significant trend toward reduced mortality (HR = 0.80, *p* = 0.238).

Assessment of the proportional hazards assumption using Schoenfeld residuals indicated no significant violation (global test: χ^2^ = 7.74, df = 4, *p* = 0.10), supporting the validity of the Cox model.

Subsequently, Kaplan–Meier analysis revealed no significant differences in survival among dogs with hemangiosarcoma when stratified by body size (Large, Medium, Small) and breeds (χ^2^ = 1.20, df = 2, *p* = 0.55; χ^2^ (15) = 17.42, *p* = 0.2942).

To further investigate survival differences among dogs diagnosed with HSA, we assessed whether clinical stage at diagnosis (T1: confined tumor; T2: evidence of rupture with consequent hemoperitoneum and emergency surgery) was associated with overall survival. Kaplan–Meier analysis and the log-rank test revealed a highly significant difference in survival distributions between the two groups (χ^2^ = 35.64, *p* < 0.001) (Figure 5). Consistently, univariable Cox proportional hazards regression confirmed that stage T2 was associated with significantly reduced survival. Compared with T1 stage, dogs with T2 tumors had more than a twofold increased hazard of death (HR = 3.17, 95% CI: 2.13–4.73, *p* < 0.001). The proportional hazards assumption was violated, as indicated by the global Schoenfeld residuals test (χ^2^ = 12.85, *p* = 0.0003). This suggests that the hazard ratio between T1 and T2 did not remain constant over time.

### 3.3. Univariate and Multivariate Logistic Regression Models

The association between various demographic and clinical factors and the likelihood of HSA was evaluated using univariate and multivariate logistic regression models (Table 4). Age of diagnosis was a significant predictor of HSA. In univariate analysis, dogs aged 6–10 years had increased odds of HSA compared to the reference group (0–5 years) (OR = 7.05, 95% CI: 1.67–29.97, *p* = 0.008). Dogs aged 11–15 years also had significantly higher odds (OR = 6.00, 95% CI: 1.41–25.39, *p* = 0.015), whereas the 16–20+ age group showed no estimable association due to the absence of events. In the multivariate model, both age groups remained significant predictors of HSA. Dogs aged 6–10 years had OR = 6.71 (95% CI: 1.56–28.75, *p* = 0.010), and those aged 11–15 years had OR = 6.25 (95% CI: 1.45–26.75, *p* = 0.014).

Purebred status was associated with decreased odds of HSA in univariate analysis (OR = 0.66, 95% CI: 0.46–0.97, *p* = 0.034). However, this effect was not statistically significant in the multivariate model (OR = 0.84, 95% CI: 0.56–1.27, *p* = 0.413).

No significant association was observed between habitat (rural or other) and HSA in either univariate or multivariate models (*p* > 0.2 for all comparisons). Sex was not significantly associated with HSA in univariate or multivariate models (*p* > 0.2).

Medium and small body sizes were associated with significantly lower odds of HSA compared to large dogs. In univariate analysis, medium dogs had an OR of 0.52 (95% CI: 0.35–0.77, *p* = 0.001) and small dogs had an OR of 0.44 (95% CI: 0.26–0.73, *p* = 0.001). These associations remained significant in the multivariate model (medium: OR = 0.58, 95% CI: 0.38–0.88, *p* = 0.011; small: OR = 0.47, 95% CI: 0.27–0.80, *p* = 0.006).

The full multivariate logistic regression, including age, breed, habitat, sex, and body size, was statistically significant (LR χ^2^ (9) = 33.08, *p* = 0.0001) and explained a modest proportion of variability in HSA occurrence (Pseudo R^2^ = 0.045). Age 6–10 and 11–15 years and smaller body size were independently associated with HSA, while breed, sex, and habitat were not significant predictors in the adjusted model.

Similar to the analysis performed for the HSA, logistic regression analyses were performed to explore the association between demographic and clinical covariates and the probability of having a non-tumoral lesion (NT) (Table 5).

In the univariate models, age was significantly associated with NT for two categories: dogs aged 6–10 years had lower odds of NT compared with the 0–5 years group (OR = 0.39, 95% CI: 0.19–0.82, *p* = 0.013), and dogs aged 11–15 years also showed significantly reduced odds (OR = 0.42, 95% CI: 0.21–0.88, *p* = 0.021). Dogs in the 16–20+ age class showed higher odds of NT, although this was not statistically significant (OR = 3.24, 95% CI: 0.63–16.31, *p* = 0.157).

Sex was significantly associated with the outcome, with males presenting higher odds of NT compared to females (OR = 1.40, 95% CI: 1.04–1.90, *p* = 0.028). Breed (purebred vs. mixed), habitat, and body size were not significantly associated with NT in univariate analysis, although medium-sized dogs approached significance (OR = 1.39, 95% CI: 0.99–1.96, *p* = 0.060). In the multivariate model, which included all covariates, both age groups 6–10 years and 11–15 years remained significantly associated with lower odds of NT (OR = 0.37, 95% CI: 0.18–0.79, *p* = 0.010 and OR = 0.40, 95% CI: 0.19–0.83, *p* = 0.015, respectively). The 16–20+ group showed increased odds but was not statistically significant (OR = 3.08, 95% CI: 0.60–15.79, *p* = 0.177). Male sex was confirmed as a significant predictor (OR = 1.48, 95% CI: 1.08–2.01, *p* = 0.014). Medium and small body size were not significantly associated with NT after adjustment, although medium-sized dogs again showed a trend (OR = 1.42, 95% CI: 0.99–2.05, *p* = 0.053). Breed type and habitat remained unrelated to NT in the adjusted model.

## 4. Discussion

This retrospective investigation, based on the analysis of 682 canine splenic lesions histopathologically diagnosed within the canine cancer registry (CCR) of central Italy (Umbria Region) between 2014 and 2023, provides a relevant contribution to the understanding of epidemiology, distribution, and prognosis of splenic diseases in dogs, with particular attention to the distinction between neoplastic and non-neoplastic lesions. Several findings of the present study are consistent with previous literature and further consolidate current knowledge, while others provide updated or underexplored insights within a contemporary, large-scale cohort.

The data collected indicated that most splenic lesions in dogs were non-neoplastic (54.2%), consistent with what has been reported in other studies [4,22,23]. The most frequent lesions were nodular hyperplasia and hematoma, changes often associated with reactive or degenerative processes rather than neoplastic processes, as already reported in the literature [4]. The overall prevalence of non-neoplastic and neoplastic diseases was comparable to prior reports, with a predominance of non-neoplastic lesions with percentages ranging from 35% to 67.5% [3,4,11,16,24,25,26]. Regarding primary splenic lymphoma, our results differ from those reported by other authors [4,16,24] but are consistent with the work of Spangler (1992) [11], who analyzed a large dataset of splenic lesions (1480 cases) in dogs.

The prevalence of lesions was higher in older dogs, peaking in the 11–15-year-old range, consistent with the reactive or neoplastic nature of many of the observed changes as previously indicated by Ko et al. [16]. Regarding breeds, purebred dogs were more frequently found to have splenic lesions than mixed breeds, but the proportion of benign tumors showed no significant differences between the two groups. Regarding the relative risks, compared to mixed breed dogs, the Lagotto Romagnolo, Boxer, Labrador Retriever and German Shepherd showed a two- to three-fold increased risk of developing HSA. Among these breeds, Boxer, Labrador Retriever and German Shepherd are well known to be predisposed [8,27], while Lagotto Romagnolo emerged as a potential new breed predisposition. However, the high prevalence of these breeds in the general population should also be considered, as it may act as a confounding factor.

Survival analysis confirmed a significantly worse prognosis of neoplastic compared to non-neoplastic lesions, as well as in individuals with HSA compared to those with non-neoplastic lesions and the other neoplastic lesions investigated in this study. Tumor rupture and hemoperitoneum represent acute life-threatening events and are associated with signs of collapse and hypovolemic shock. Furthermore, clinical staging within hemangiosarcoma cases was strongly associated with survival, confirming the aggressive biological behavior of this tumor as already highlighted by [28].

The association between demographic and clinical variables and the likelihood of HSA was further explored using univariate and multivariate logistic regression models. Age at diagnosis was a significant predictor: dogs aged 6–10 years had markedly increased odds of HSA compared to the reference group (0–5 years) in both univariate (OR = 7.05, 95% CI: 1.67–29.97, *p* = 0.008) and multivariate analyses (OR = 6.71, 95% CI: 1.56–28.75, *p* = 0.010). Similarly, dogs aged 11–15 years showed significantly higher odds of HSA (univariate OR = 6.00, 95% CI: 1.41–25.39, *p* = 0.015; multivariate OR = 6.25, 95% CI: 1.45–26.75, *p* = 0.014). These results are consistent with previous reports, as HSA predominantly affects older dogs, with a median onset at 10 years [29,30]. Similarly, Carnio and colleagues, analyzing risk factors for canine visceral HSA, reported a markedly increased susceptibility (OR = 14.01, 95% CI: 1.65–119.03) for dogs older than 10 years compared to the reference age group of 0–5 years [31].

Purebred status was associated with decreased odds in univariate analysis (OR = 0.66, 95% CI: 0.46–0.97, *p* = 0.034), but not in the adjusted model (OR = 0.84, 95% CI: 0.56–1.27, *p* = 0.43). Sex and habitat were not significantly associated with HSA in either univariate or multivariate models. Medium and small body sizes, accordingly with Sherwood and colleagues, were associated with lower odds compared to large dogs, and these effects remained significant in the multivariate model (medium: OR = 0.58, 95% CI: 0.38–0.88, *p* = 0.011; small: OR = 0.47, 95% CI: 0.27–0.80, *p* = 0.006) [25]. Importantly, the multivariate approach adopted in the present study allowed a refined evaluation of previously reported associations by jointly considering covariates that had been analyzed separately in earlier studies. While age and body size emerged as independent predictors of hemangiosarcoma risk, other variables lost statistical significance after adjustment, underscoring the importance of integrating multiple demographic and environmental factors within a single analytical framework.

Overall, survival and multivariate analysis showed that diagnosis of HSA, advanced age and body size were the main risk factors for reduced survival. The lack of significant associations between other demographic variables (sex, breed, habitat) and prognosis emphasizes that histological diagnosis remains the primary determinant of clinical outcome in dogs undergoing splenectomy.

Logistic regression models were also used to evaluate predictors of non-tumoral lesions. Male sex was a significant predictor of non-tumoral lesions (OR = 1.48, 95% CI: 1.08–2.01, *p* = 0.014), suggesting that, in our cohort, male dogs were more likely to present with non-neoplastic splenic lesions. This contrasts with previous reports, where male sex has been consistently associated with an increased risk of splenic tumoral nodular lesions [4]. This discrepancy may reflect differences in study populations, diagnostic inclusion criteria, or geographic variations, and suggests that sex-related risk profiles for splenic lesions may be more complex than previously assumed.

A comparison with the largest previously published cohort on canine splenic diseases further contextualizes the present findings. The study by Spangler (1992), based on 1480 cases collected between 1985 and 1989, provided a foundational description of splenic lesion distribution in dogs [11]. However, that historical cohort predates current histopathological classification systems, advances in diagnostic imaging, and modern clinical management of splenic disease. In contrast, the present study analyzes a contemporary population (2014–2023) with standardized histopathological diagnoses and integrates multivariate risk modeling and survival analysis, approaches that were not routinely applied in earlier large-scale investigations. These methodological and temporal differences support the need to reassess previously reported epidemiological patterns within a modern clinical context rather than relying exclusively on historical data.

From a clinical perspective, the present findings provide updated evidence to support diagnostic and prognostic decision-making in dogs undergoing splenectomy. The high proportion of non-neoplastic lesions reinforces the limited predictive value of demographic variables alone and underscores the central role of histopathological examination in guiding prognosis. Updated risk estimates for age and body size may assist clinicians in preoperative counseling and postoperative management, particularly in older and large-breed dogs.

Despite the data so far presented, some potential limitations must be considered for this study; for instance, the database used, although extensive, does not include detailed information on preoperative clinical status, post-surgical treatment, or the presence of metastases at the time of diagnosis, factors that could significantly influence the prognosis, particularly for the HSA.

## 5. Conclusions

The results of this study highlight the pivotal role of histopathologic evaluation in distinguishing canine splenic HSA from other lesions, confirming it as the most clinically relevant and aggressive neoplasm among splenic tumors in dogs. Accurate histologic diagnosis remains essential to guide appropriate clinical management and improve the clinical management and outcome of patients. HSA continues to pose a significant clinical challenge, with advanced age, larger body size, and higher clinical stage (T2) being associated with poorer outcomes. Logistic regression analyses highlighted that dogs aged 6–10 and 11–15 years and those of large body size are at higher risk of HSA, whereas other demographic factors, such as breed, sex, and habitat, were not independently predictive. These findings provide updated epidemiologic insights into the distribution and risk factors of canine splenic lesions, supporting evidence-based clinical assessment and population-level understanding of this disease.

## Figures and Tables

**Figure 1 vetsci-13-00064-f001:**
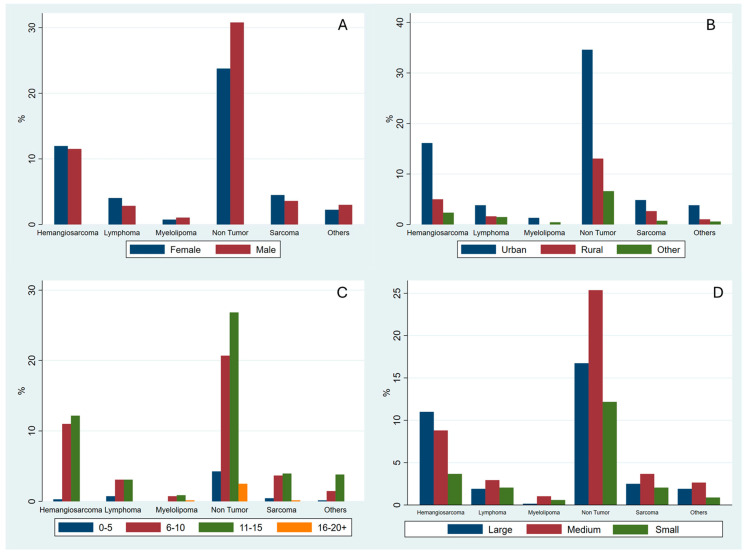
Distribution of histological diagnoses stratified by sex (**A**), geographical area (**B**), age group (**C**), and size (**D**).

**Figure 2 vetsci-13-00064-f002:**
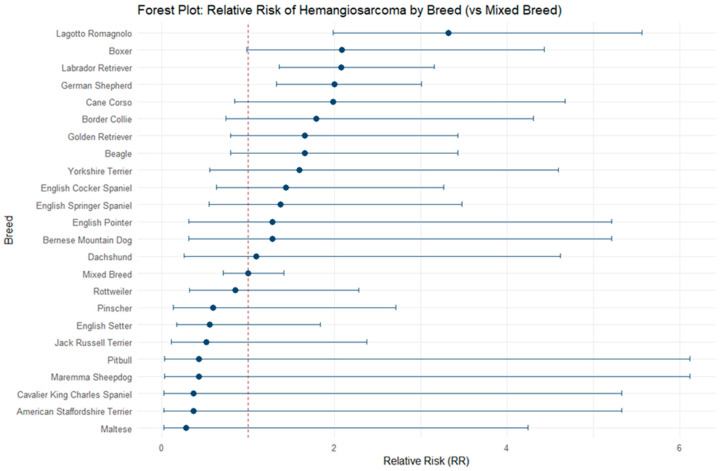
Forest plot reporting relative risks (RRs) of HSA development in different dog breeds compared to mixed-breed dogs.

**Figure 3 vetsci-13-00064-f003:**
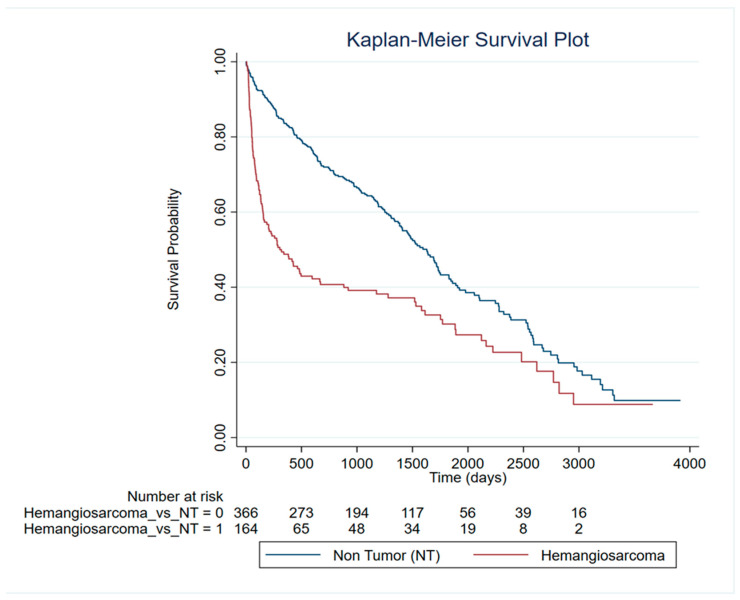
Survival analysis of HSA patients compared to non-tumor dogs.

**Figure 4 vetsci-13-00064-f004:**
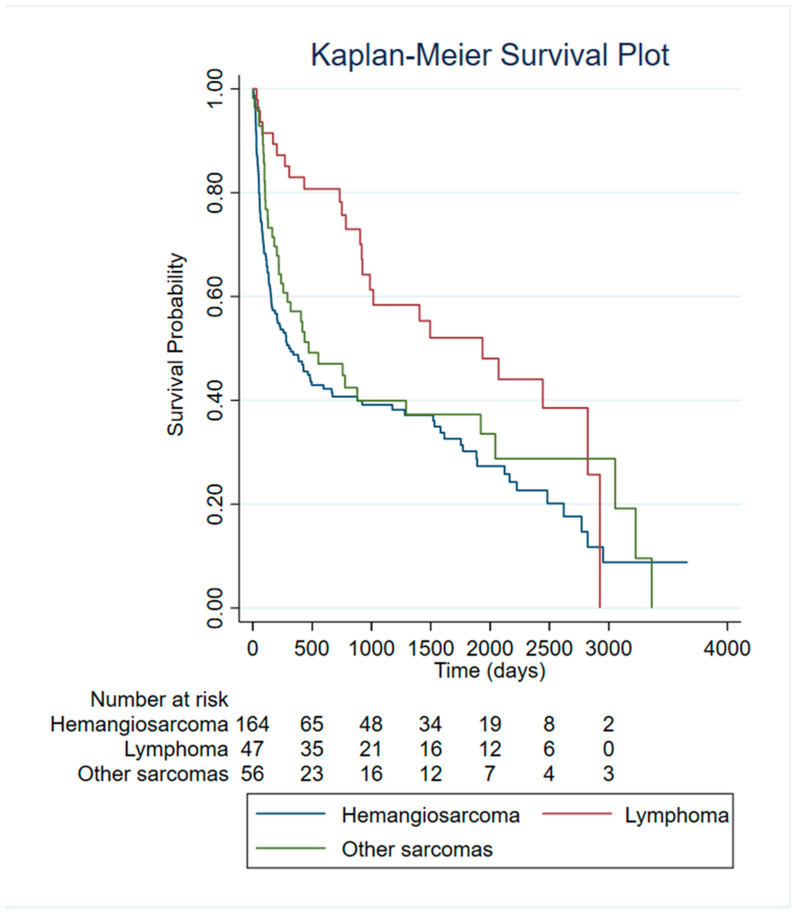
Survival analysis of HSA patients compared to other malignant tumors.

**Figure 5 vetsci-13-00064-f005:**
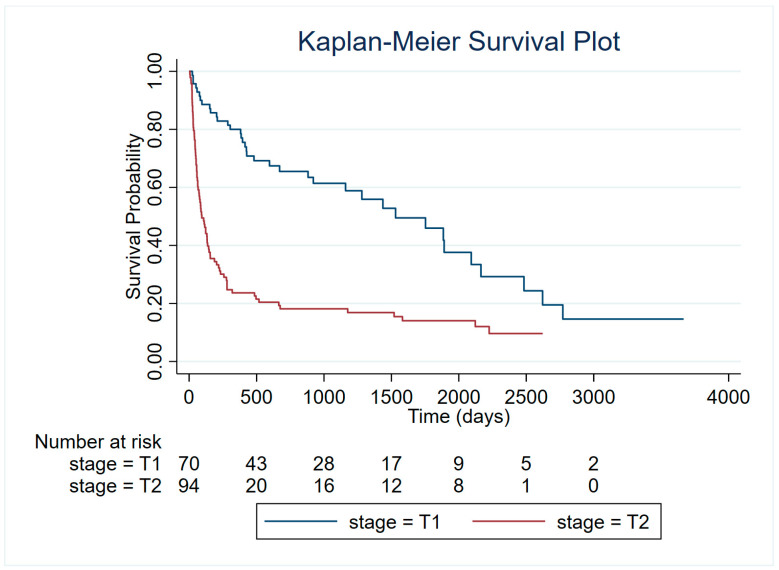
Kaplan–Meier survival curves for dogs with HSA stratified by clinical stage at diagnosis (T1: confined tumor; T2: evidence of rupture).

**Table 1 vetsci-13-00064-t001:** Distribution of canine splenic lesions according to breed, sex, and age.

Variables	Splenic Lesions	Non-Neoplastic	Neoplastic
	n (%)	n (%)	n (%)
**Breed**			
Mixed	262 (38.4)	140 (37.8)	122 (39.1)
Pure	420 (61.6)	230 (62.2)	190 (60.9)
Total	682 (100)	370 (100)	312 (100)
**Sex**			
Female	325 (47.6)	162 (43.8)	163 (52.2)
Male	357 (52.4)	208 (56.2)	149 (47.8)
Total	682 (100)	370 (100)	312 (100)
**Age**			
0–5 years	40 (5.9)	29 (7.8)	11 (3.5)
6–10 years	277 (40.6)	141 (38.1)	136 (43.6)
11–15 years	346 (49.3)	183 (49.4)	163 (52.3)
16–20+ years	19 (2.8)	17 (4.6)	2 (0.6)
Total	682 (100)	271 (100)	411 (100)

**Table 2 vetsci-13-00064-t002:** Frequency of non-neoplastic and neoplastic lesions according to histopathology.

Histopathological Diagnosis	n	%
Non-neoplastic	370	54.3
Hematoma	211	57.0
Nodular hyperplasia	138	37.3
Splenitis	18	4.9
Trombosis	3	0.8
Neoplastic	312	45.7
Hemangiosarcoma	170	54.5
Other sarcomas	59	19.0
Lymphoma	47	15.1
Metastatic carcinoma	4	1.3
Myelolipoma	12	3.8
Hemangioma	9	2.9
Lipoma	3	1.0
Extraskeletal osteosarcoma	2	0.6
Plasmacytoma	2	0.6
Leiomyoma	1	0.3
Lymphangioma	1	0.3
Metastatic undifferentiated neoplasia	1	0.3
Malignant mesenchymoma	1	0.3
TOTAL	682	100

**Table 3 vetsci-13-00064-t003:** Relative risk (RR) of HSA development in different dog breeds compared to mixed-breed dogs.

Breed	RR	Lower CI	Upper CI
Lagotto Romagnolo	3.32	1.98	5.56
Boxer	2.09	0.99	4.43
Labrador Retriever	2.07	1.36	3.16
German Shepherd	2.00	1.32	3.01
Cane Corso (Italian mastiff)	1.99	0.84	4.67
Border Collie	1.79	0.74	4.31
Beagle	1.65	0.80	3.43
Golden Retriever	1.65	0.80	3.43
Yorkshire Terrier	1.60	0.55	4.59
English Cocker Spaniel	1.44	0.63	3,26
English Springer Spaniel	1.37	0.54	3.48
Bernese Mountain Dog	1.28	0.31	5.22
English Pointer	1.28	0.31	5.22
Dachshund	1.09	0.26	4.62
Mixed Breed	1.00	0.71	1.41
Rottweiler	0.85	0.32	2.28
Pinscher	0.59	0.13	2.71
English Setter	0.56	0.17	1.83
Jack Russell Terrier	0.51	0.11	2.38
Maremma Sheepdog	0.43	0.03	6.12
Pitbull	0.43	0.03	6.12
American Staffordshire Terrier	0.36	0.02	5.33
Cavalier King Charles Spaniel	0.36	0.02	5.33
Maltese	0.28	0.02	4.24

**Table 4 vetsci-13-00064-t004:** Univariate and multivariate logistic regression analysis of demographic factors associated with HSA. Odds ratios (OR), 95% confidence intervals (CI), and *p*-values are reported. Reference categories are indicated for each variable.

Predictor	Category	Reference	OR (Univariate)	95% CI	*p*-Value	OR (Multivariate)	95% CI	*p*-Value
**Age group**	6–10	0–5	7.05	1.67–29.97	0.008	6.71	1.56–28.75	0.010
	11–15	0–5	6.00	1.41–25.39	0.015	6.25	1.45–26.75	0.014
	16–20+	0–5	1	—	—	—	—	—
**Breed**	Purebred	Mixed	0.66	0.46–0.97	0.034	0.84	0.56–1.27	0.413
**Habitat**	Rural	Urban	0.82	0.53–1.26	0.361	0.82	0.52–1.29	0.398
	Other	Urban	0.72	0.40–1.29	0.265	0.74	0.41–1.37	0.340
**Sex**	Male	Female	0.83	0.58–1.18	0.298	0.77	0.53–1.11	0.160
**Body size**	Medium	Large	0.52	0.35–0.77	0.001	0.58	0.38–0.88	0.011
	Small	Large	0.44	0.26–0.73	0.001	0.47	0.27–0.80	0.006

**Table 5 vetsci-13-00064-t005:** Univariate and multivariate logistic regression analysis of demographic factors associated with non-tumoral splenic lesions. Odds ratios (OR), 95% confidence intervals (CI), and *p*-values are reported. Reference categories are indicated for each variable.

Predictor	Category	Reference	OR (Univariate)	95% CI	*p*-Value	OR (Multivariate)	95% CI	*p*-Value
**Age group**	6–10	0–5	0.39	0.19–0.82	0.013	0.37	0.18–0.79	0.010
	11–15	0–5	0.42	0.21–0.88	0.021	0.40	0.19–0.83	0.015
	16–20+	0–5	3.24	0.63–16.31	0.157	3.08	0.60–15.79	0.177
**Breed**	Purebred	Mixed	0.95	0.70–1.29	0.735	0.82	0.58–1.15	0.242
**Habitat**	Rural	Urban	1.10	0.76–1.58	0.612	1.11	0.76–1.63	0.590
	Other	Urban	1.02	0.64–1.64	0.922	0.97	0.60–1.59	0.910
**Sex**	Male	Female	1.40	1.04–1.90	0.028	1.48	1.08–2.01	0.014
**Body size**	Medium	Large	1.39	0.99–1.96	0.060	1.42	0.99–2.05	0.053
	Small	Large	1.38	0.91–2.09	0.134	1.39	0.89–2.17	0.149

## Data Availability

Data is unavailable due to privacy or ethical restrictions. For further information, write to e.manuali@izsum.it.

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
