# Peer review of "Epidemiology of Splenic Lesions in Dogs Undergoing Splenectomy—Pathological Characterization and Risk Factors"

_vetsci, 2026, doi:10.3390/vetsci13010064_

Round 1
Reviewer 1 Report
Comments and Suggestions for Authors
In this report titled “Epidemiology of splenic lesions in dogs undergoing splenectomy. Pathological characterization and risk factors”, the authors conducted an epidemiological study of data on splenic lesions in dogs in a specific region of Italy. Unfortunately, as the authors state in this report, it appears that no new insights were provided for veterinarians involved in the diagnosis and treatment of canine splenic lesions. More specifically, most of the findings obtained in this study are consistent with those reported in previous studies. Additionally, while breed involvement is mentioned, confounding factors have not been sufficiently examined, and the details remain unclear. Based on the above, I believe this report does not go beyond merely replicating previously reported research.
Specific comments
>Introduction
The majority of the composition consists of a mere listing of previously reported paper information, making it unclear what issues the authors consider important in this study. If this is simply a study that changed the exploration area, it could be considered a study of little value to many clinical veterinarians.
>Results
Although enormous amounts of analysis have been conducted, the purpose of each analysis is unclear. A more detailed chapter structure and consideration for the readers would be necessary.
Most tables span multiple pages, making it very difficult to review.
>Discussion
The majority consists of repeated results. Furthermore, most of these are consistent with previously reported findings, making the novelty of this study unclear.
Author Response
Specific comments
>Introduction
The majority of the composition consists of a mere listing of previously reported paper information, making it unclear what issues the authors consider important in this study. If this is simply a study that changed the exploration area, it could be considered a study of little value to many clinical veterinarians.
Author’s response: We appreciate the reviewer’s comment. While we understand the concern that the manuscript’s introduction could appear to be a descriptive aggregation of prior knowledge, this was not mentioned by the other three reviewers and seems quite smooth to the authors and needed to prepare any reader, even the one completely lacking knowledge on the topic to the manuscript. Regarding the statement “If this is simply a study that changed the exploration area, it could be considered a study of little value to many clinical veterinarians”; Our study examines 682 canine spleens, a substantially larger cohort than all of the most frequently cited studies in this field with datasets ranging from 65 (Ref. Mass-to-splenic volume ratio and splenic weight as a percentage of body weight; Mallinckrodt and Gottfried, J Am Vet Med Assoc. 2011) to 507 reports (Ref. Pathological Characterization and Risk Factors of Splenic Nodular Lesions in Dogs (Canis lupus familiaris); Corvera et al. Animals, 2024).
Compared with these, our data set represents the largest contemporary cohorts evaluating splenic lesions in dogs. This expanded sample size allows us to strengthen, refine, or challenge diagnostic assumptions made in smaller studies and provides more robust estimates of lesion prevalence across a diverse population. Importantly, our objective is not merely to describe our findings but to evaluate how these results align with or differ from existing literature and related to potential risk and predisposition factors as well as prognostic data. Additionally, we would like to emphasize that one of the key contributions of our work is to expand high-quality primary data freely available to the field, thereby supporting future systematic reviews and meta-analyses, which require large amount of data to generate meaningful conclusions; our work goes against the grain of the ‘first arrive, first publish’ novelty logic, where the first study tends to automatically become the truth, and instead emphasizes the importance of broader, more robust studies that can provide better data for future analysis and truly reliable clinical support.
A clarification on the aims and on the point of strength of this study has been provided in the introduction (LINE 95-102)
We hope this clarification demonstrates the value and relevance of our study to the veterinary community and addresses the reviewer’s concerns.
>Results
Although enormous amounts of analysis have been conducted, the purpose of each analysis is unclear. A more detailed chapter structure and consideration for the readers would be necessary.
Author’s response: We appreciate the opportunity to clarify the purpose and rationale of each analytical component conducted in the study. Although the manuscript contains several layers of analysis, each served a specific and sequential purpose. First the study population and the distribution of splenic lesion types were described (e.g., frequencies) for a general overview of the dataset; then breed- and size and other specific patterns and estimated relative risks were investigated for hemangiosarcoma and non-neoplastic lesions according to demographic characteristics. For HSA, the main concern when considering canine splenic masses, survival outcomes were evaluated using Kaplan–Meier analysis and Cox regression models. Finally, univariate and multivariate logistic regression analyses were performed to identify demographic predictors of hemangiosarcoma.
A clarification on the purpose of each analysis of this study has been provided at the beginning of each paragraph (LINE 193-194, 211-212, 226-227, 267-269, 278-279, 321-322)
We hope this clarification improves the readability and clarifies the purpose of each analysis of this study.
Most tables span multiple pages, making it very difficult to review.
Author’s response: We appreciate the reviewer’s comment. However, this was related to the line spacing option, previously set at 2,0 to give higher readability for revision. These are now set at 1, much compacted, as they should appear in the final version.
>Discussion
The majority consists of repeated results. Furthermore, most of these are consistent with previously reported findings, making the novelty of this study unclear.
Author’s response: We appreciate the reviewer’s comment that gives us the opportunity to clarify that, despite some overlap with prior observations, our analysis of 682 canine spleens offers substantially greater statistical power than earlier studies and provides novel insights into demographic risk patterns and lesion distribution that were not previously detectable in smaller cohorts.

Reviewer 2 Report
Comments and Suggestions for Authors
-

Author Response
Comments: It is unclear whether the authors performed the experiments leading to the histopathological and immunochemical diagnosis or simply described the techniques used by others (data collection section). If these data are the authors' own, they should present representative histochemical and immunohistochemical results; otherwise, they should identify the laboratory that performed them and add a reference.
Author’s response: We thank the reviewer for this crucial observation. All histopathological and immunohistochemical diagnoses included in this study derive from our canine cancer registry (CCR) of central Italy (Umbria Region) between 2014 and 2023, so these are the authors' own data. The authors would like to specify that histochemical and immunohistochemical results were performed at time of diagnosis for each case and not specifically for this study. Because the study involved only retrospective data collection of previously issued pathology reports, and not the generation of new diagnostic material, representative histochemical or immunohistochemical images were not included as part of this work. Regarding the diagnosis: our diagnostic system requires that each case is independently evaluated by two pathologists. A diagnosis is issued and sent to the referring veterinarian (and subsequently automatically registered in the Registry) only when the two pathologists reach full agreement. In cases of disagreement, a third pathologist blindly reviews the slides to resolve the discrepancy. This ensures high diagnostic accuracy and consistency across the dataset and limits the misdiagnoses improving quality for the clients and patient care for the dogs.
Minor changes have been made to the manuscript to clarify these elements (LINE 107, 135-140, 158-162)
We trust this clarification will resolve the concern and make the diagnostic procedures underlying our dataset more transparent.
The article is very repetitive in its presentation of the data obtained. It shows and mentions them in the text, tables, and figues, making for an uninteresting read. Avoid such repetitiveness. This applies to data presented in the text, figures 1-2, and tables 1-2. Select the most appropriate way to outline the results, either in figures or tables, and mention the most notable aspects in the text section.
However, table 3 and figure 4, as well as figure 4, highlight the analysis performed.
Author’s response: We appreciate the reviewer’s comment. We agree that reducing redundancy improves readability and strengthens the presentation of the results. To address this, Figure 1 has been removed, as its content overlapped with information already clearly presented in Table 1. Table 1 has been retained, for the reasons mentioned. Figure 2 and Table 2 have both been kept, because they present complementary rather than redundant information: Table 2, reports only the absolute (n) and relative (%) frequencies of each lesion type (neoplastic and non-neoplastic) without stratification, providing a “pure” baseline distribution that cannot be derived directly from Figure 2. Figure 2, in contrast, displays the distribution of lesion types stratified by sex, habitat, age, and size, offering a visual synthesis of patterns across demographic groups.
The graph in figure 4 says "Emangiosarcoma." Please correct it.There are some typographical errors that need to be corrected
Author’s response: We thank the reviewer for his suggestion. Figure has been amended as required.

Reviewer 3 Report
Comments and Suggestions for Authors
As the authors mentioned, this study provides important insights into the epidemiologic distribution and risk factors of canine splenic lesions through the analysis of 682 cases over a 10-year period.
However, the method by which the extensive dataset was organized is not clearly described, and the presentation of the results is not well structured, causing confusion for the reader. The overall writing quality is somewhat lacking, and I believe the manuscript requires careful revision by the authors. In addition to the minor comments I will point out regarding grammar, spacing, and consistency of terminology, I strongly recommend thorough proofreading for refinement and coherence.
Major comments)
- Line 57: It is inappropriate to describe 2/3 as “most.”
- Lines 106 and 133: It is stated that the tissue samples were processed at the time of submission. However, were all histopathologic samples of surgically resected canine spleens re-reviewed by three pathologists, and were additional stains performed if necessary? I do not fully understand the methodology as described. Please clarify whether this study retrospectively reviewed the histopathologic results from the time of initial diagnosis in 682 cases, or whether the paraffin blocks were re-examined microscopically.
- Line 126: It is stated that all dogs included in the study had follow-up data. Were dogs without follow-up data excluded from this study? In addition, it appears that not all cases were included in the survival analysis (e.g., in the HSA and no-tumor groups). Please clarify this point.
[Results]
- Much of the text in the Results section duplicates data already shown in figures or tables. There is no need to list the same values twice. Please organize the results clearly to avoid confusion for readers.
- The descriptions are inconsistent—for example, (cases, %), (n/N, %), and mixed usage of “case,” “subject,” and “dog.” Please unify these expressions and present the data in tables where possible for better readability, describing only the essential results in the text.
- Figure 1A,B are unnecessary; a table would suffice.
- Table 1: the age groupings differ from those in the Introduction and Figure 2. Unless there is a specific reason for this, please standardize the grouping criteria.
- Line 322: What does “non-NT lesions” refer to? Also, as mentioned earlier, why are the numbers of cases inconsistent? Among the 170 dogs with HSA, were 60 still alive, or were they lost to follow-up? If follow-up data were unavailable, the Methods section should be revised accordingly.
- Figure 4: What does the “number at risk” shown below the x-axis represent? Is this information essential in this figure? The same question applies to Figures 5 and 6. If this element is necessary, it should be clearly explained in the figure legend.
[Discussion]
- Line 456: Does this statement refer to your own study results, or to those from references 4, 22, and 23? In your histopathologic findings, splenic congestion was not mentioned.
- Line 469: In my opinion, the discussion related to sex is unnecessary or inconsistent with the surrounding context.
- Lines 488–491: This section merely repeats the results without adding new interpretation.
- Line 496: What was the incidence of rupture or hemoperitoneum among dogs diagnosed with hemangiosarcoma in your study? Since you mentioned that follow-up data were available for all dogs, this information is expected to be included.
- Lines 504–526: The numerical data already presented in both the results section and tables are redundantly repeated here. Only essential findings should be restated, with greater focus placed on interpretation and discussion. In addition, avoid repeatedly using expressions such as “Carnio and colleagues” or “Sherwood and collaborators” when citing references.
- Line 544: You mentioned as a study limitation that the database does not include detailed information on preoperative clinical status, post-surgical treatment, or the presence of metastases at the time of diagnosis—factors that could significantly influence prognosis. However, it seems likely that such information was available for a considerable number of the 682 cases. Additional discussion related to these aspects would make your study much more meaningful.
- Your study includes not only hemangiosarcoma but also a considerable number of other malignant tumors, such as other sarcomas and lymphomas. I recommend adding a discussion on the proportion of benign versus malignant neoplastic lesions, as well as on the differences in lymphoma incidence compared to previous studies.
Minor comments)
- There are multiple spacing errors throughout your manuscript, particularly in the citation numbering.
- Line 117: The abbreviation for overall survival (OS) is unnecessary.
- Line 148: Superscript formatting issue.
- Line 206: What does “DS” refer to? Is it the same as “SD” on line 202, which stands for standard deviation (line 208)? Please clarify precisely.
- Line 220: You use both “HSA” and “hemangiosarcoma” interchangeably. Please maintain consistency.
- Table 3: The numbers should use a period instead of a comma as a decimal separator, according to standard formatting.
- Figures: Ensure that “Figure” and “Fig.” usage is consistent throughout and follows the journal’s guidelines.
- Line 319: Are “neoplastic” and “tumor” referring to the same concept? Please clarify.
- Figure 4: In the legend, there is inconsistency between “Tumor” and “Tumors.”
- Line 379: When dividing by age, what is the difference between “group” and “class”? Please clarify.
Comments on the Quality of English LanguageThe English language throughout the manuscript is generally understandable, but the writing is occasionally disorganized and unpolished. Careful proofreading and editing are recommended to improve clarity, consistency, and overall readability.
Author Response
Comments and Suggestions for Authors
As the authors mentioned, this study provides important insights into the epidemiologic distribution and risk factors of canine splenic lesions through the analysis of 682 cases over a 10-year period.
However, the method by which the extensive dataset was organized is not clearly described, and the presentation of the results is not well structured, causing confusion for the reader. The overall writing quality is somewhat lacking, and I believe the manuscript requires careful revision by the authors. In addition to the minor comments I will point out regarding grammar, spacing, and consistency of terminology, I strongly recommend thorough proofreading for refinement and coherence.
Major comments)
Author’s response: We thank the reviewer for this constructive comment. In response, we have introduced clear incipit sentences at the beginning of each analytical subsection to improve structure and guide the reader through the logic of the analyses (LINE 193-194, 211-212, 226-227, 267-269, 278-279, 321-322). We have also clarified in the Methods that the dataset was obtained retrospectively from the Tumor Registry, where we collected only the diagnostic and demographic information already present in the archive (LINE 107, 135-140, 158-162). These revisions were aimed at improving clarity and readability while accurately reflecting how the data were handled in this study.
- Line 57: It is inappropriate to describe 2/3 as “most.”
Author’s response: We appreciate the reviewer’s comment. This has been rephrased as: “…splenic mass lesions are more frequently benign” (LINES 60-61).
- Lines 106 and 133: It is stated that the tissue samples were processed at the time of submission. However, were all histopathologic samples of surgically resected canine spleens re-reviewed by three pathologists, and were additional stains performed if necessary? I do not fully understand the methodology as described. Please clarify whether this study retrospectively reviewed the histopathologic results from the initial diagnosis in 682 cases, or whether the paraffin blocks were re-examined microscopically.
Author’s response: We thank the reviewer for this crucial observation. All histopathological and immunohistochemical diagnoses included in this study derive from our canine cancer registry (CCR) of central Italy (Umbria Region) between 2014 and 2023, so these are the authors' own data. The authors would like to specify that histochemical and immunohistochemical results were performed at time of diagnosis for each case and not specifically for this study. Because the study involved only retrospective data collection of previously issued pathology reports, and not the generation of new diagnostic material, representative histochemical or immunohistochemical images were not included as part of this work. Regarding the diagnosis: our diagnostic system requires that each case is independently evaluated by two pathologists. A diagnosis is issued and sent to the referring veterinarian (and subsequently automatically registered in the Registry) only when the two pathologists reach full agreement. In cases of disagreement, a third pathologist blindly reviews the slides to resolve the discrepancy. This ensures high diagnostic accuracy and consistency across the dataset and limits the misdiagnoses improving quality for the clients and patient care for the dogs.
Minor changes have been made to the manuscript to clarify these elements (LINES 107-110, 133-140, 158-162)
We trust this clarification will resolve the concern and makes the diagnostic procedures underlying our dataset more transparent.
- Line 126: It is stated that all dogs included in the study had follow-up data. Were dogs without follow-up data excluded from this study? In addition, it appears that not all cases were included in the survival analysis (e.g., in the HSA and no-tumor groups). Please clarify this point.
Author’s response: We thank the reviewer for this crucial observation. All dogs have available follow-up data. Due to a software code the first analysis failed to include some cases in the analysis. This has now been amended. Anyway, 10 dogs were not included in the survival analysis since their diagnosis and death dates coincided.
[Results]
- Much of the text in the Results section duplicates data already shown in figures or tables. There is no need to list the same values twice. Please organize the results clearly to avoid confusion for readers.
Author’s response: Thanks for the suggestions. We removed Figure 1a and 1b
- The descriptions are inconsistent—for example, (cases, %), (n/N, %), and mixed usage of “case,” “subject,” and “dog.” Please unify these expressions and present the data in tables where possible for better readability, describing only the essential results in the text.
Author’s response: Thanks for the suggestions. We have aligned the terms in the text and deleted redundant sentences.
- Figure 1A,B are unnecessary; a table would suffice.
Author’s response: We appreciate the reviewer’s comment. These have been removed. Related Information are in Table 1, 2.
- Table 1: the age groupings differ from those in the Introduction and Figure 2. Unless there is a specific reason for this, please standardize the grouping criteria.
Author’s response: We thank the reviewer for this comment. The table has been corrected and text amended.
- Line 322: What does “non-NT lesions” refer to? Also, as mentioned earlier, why are the numbers of cases inconsistent? Among the 170 dogs with HSA, were 60 still alive, or were they lost to follow-up? If follow-up data were unavailable, the Methods section should be revised accordingly.
Author’s response: We thank the reviewer for this comment. In the manuscript the term NT refers to non-tumoral lesions, i.e., non-neoplastic conditions. The expression ‘non-NT lesions’ was unintentionally ambiguous and has now been replaced with the clearer term ‘non-neoplastic lesions’.
Regarding case numbers: all 170 dogs diagnosed with HSA had available follow-up data. Among them, 6 dogs presented the same diagnosis and death dates. No dogs were lost to follow-up.
- Figure 4: What does the “number at risk” shown below the x-axis represent? Is this information essential in this figure? The same question applies to Figures 5 and 6. If this element is necessary, it should be clearly explained in the figure legend.
Author’s response: We appreciate the reviewer’s comment. While we understand that this information might be unclear for the reviewer. The “number at risk” shown below the x-axis in Figures 4, 5, and 6 represents the number of subjects still at risk (patients that have not experienced the event (death); basically, dogs that are still alive in that specific time point) in each time interval of the survival curve. This is standard information included in Kaplan–Meier survival analyses, useful and needed for interpreting the robustness of survival estimates as the number of subjects observed decreases. Due to the importance of this date, we have decided to keep the figures as they are in the current state. We hope this clarification addresses the reviewer’s concerns.
[Discussion]
- Line 456: Does this statement refer to your own study results, or to those from references 4, 22, and 23? In your histopathologic findings, splenic congestion was not mentioned.
Author’s response: We appreciate the reviewer’s comment. Splenic congestion was observed in association with other splenic lesions (e.g. hemorrhages, infarctions) and therefore was not reported on the table. The terms has been deleted to avoid misunderstanding.
- Line 469: In my opinion, the discussion related to sex is unnecessary or inconsistent with the surrounding context.
Author’s response: As required this part of discussion has been deleted.
- Lines 488–491: This section merely repeats the results without adding new interpretation.
Author’s response: We appreciate the reviewer’s comment. Lines have been removed.
- Line 496: What was the incidence of rupture or hemoperitoneum among dogs diagnosed with hemangiosarcoma in your study? Since you mentioned that follow-up data were available for all dogs, this information is expected to be included.
Author’s response: We appreciate the reviewer’s comment. As clarified in the revised text, all dogs classified as T2 at clinical staging presented with splenic rupture, resulting in hemoperitoneum and requiring emergency surgery. By definition, canine splenic hemangiosarcomas must be associated with rupture and hemoperitoneum to be classified as T2.
- Lines 504–526: The numerical data already presented in both the results section and tables are redundantly repeated here. Only essential findings should be restated, with greater focus placed on interpretation and discussion. In addition, avoid repeatedly using expressions such as “Carnio and colleagues” or “Sherwood and collaborators” when citing references.
Author’s response: We appreciate the reviewer’s comment. Some lines have been removed. The term “colleagues” has been standardized.
- Line 544: You mentioned as a study limitation that the database does not include detailed information on preoperative clinical status, post-surgical treatment, or the presence of metastases at the time of diagnosis—factors that could significantly influence prognosis. However, it seems likely that such information was available for a considerable number of the 682 cases. Additional discussion related to these aspects would make your study much more meaningful.
Author’s response: We appreciate the reviewer’s comment but we do not have such information.
- Your study includes not only hemangiosarcoma but also a considerable number of other malignant tumors, such as other sarcomas and lymphomas. I recommend adding a discussion on the proportion of benign versus malignant neoplastic lesions, as well as on the differences in lymphoma incidence compared to previous studies.
Author’s response: As suggested by the reviewer we added these lines: “The overall prevalence of non-neoplastic and neoplastic diseases was comparable to prior reports, with a predominance of non-neoplastic lesions with percentages ranging from 56 to 67.5% (Spangler 1992, O’Byrne 2019, Ko 2023, Corvera 2024 and Ziogaite 2024). Regarding primary splenic lymphoma, our results differ from those reported by other Authors (O’Birne, Corvera, Ko) but are consistent with the work of Spangler (2019), who analyzed a large dataset of splenic lesions (1,480 cases) in dogs” (Lines 465-471).
Minor comments)
- There are multiple spacing errors throughout your manuscript, particularly in the citation numbering.
Author’s response: We amended spacing errors throughout the manuscript.
- Line 117: The abbreviation for overall survival (OS) is unnecessary.
Author’s response: We amended as suggested.
- Line 148: Superscript formatting issue.
Author’s response: We amended as suggested.
- Line 206: What does “DS” refer to? Is it the same as “SD” on line 202, which stands for standard deviation (line 208)? Please clarify precisely.
Author’s response: We amended as suggested. DS stood for SD. It was a typo.
- Line 220: You use both “HSA” and “hemangiosarcoma” interchangeably. Please maintain consistency.
Author’s response: We amended as suggested by the reviewer.
- Table 3: The numbers should use a period instead of a comma as a decimal separator, according to standard formatting.
Author’s response: We amended according to standard formatting.
- Figures: Ensure that “Figure” and “Fig.” usage is consistent throughout and follows the journal’s guidelines.
Author’s response: We modified throughout the entire manuscript the term “Fig.” into “Figure”.
- Line 319: Are “neoplastic” and “tumor” referring to the same concept? Please clarify.
Author’s response: We apologize with the reviewer but searching the first version of the manuscript we did not find the line with the terms “neoplastic” and “tumor”.
- Figure 4: In the legend, there is inconsistency between “Tumor” and “Tumors.”
Author’s response: modified as required.
- Line 379: When dividing by age, what is the difference between “group” and “class”? Please clarify.
Author’s response: We used the terms “group” and “class” as synonym. We changed the term “class” into “group” to be consistent throughout the manuscript.

Reviewer 4 Report
Comments and Suggestions for Authors
The manuscript describes an interesting epidemiological analysis of dogs with neoplastic and non-neoplastic splenic lesions in a region of Italy. Comments are provided to better understand and improve the quality of some results:
Methods
- What were the criteria for establishing the 5 age ranges? Clarify
- In Figure 1, correct Figure 1B where it is described "emangiosarcoma". This term is also used in other parts of the text. Please, correct
- Describe with more detail logistic regression method
Results
- Survival curves show highly predictable and relatively uninformative results. For example, comparing hemangiosarcoma with non-tumor lesions provides no novel information. I recommend that survival analyses of animals with hemangiosarcoma be performed according to breed and size. Survival could also be compared between HSA and other tumors.
- The table under the survival curve shown in Figure 6 is not understood. This curve is also not well understood. Do patients with HSA have different survival rates depending on their age at diagnosis? Please, clarify
Author Response
Comments on the Quality of English Language
The English language throughout the manuscript is generally understandable, but the writing is occasionally disorganized and unpolished. Careful proofreading and editing are recommended to improve clarity, consistency, and overall readability.
The manuscript describes an interesting epidemiological analysis of dogs with neoplastic and non-neoplastic splenic lesions in a region of Italy. Comments are provided to better understand and improve the quality of some results:
Methods
- What were the criteria for establishing the 5 age ranges? Clarify
Author’s response: The classification adopted (0–5, 6–10, 11–15, 16–20+, years) was chosen to represent significant biological and clinical stages (young, adult, elderly, very elderly and ultra-elderly).
This subdivision is consistent with that reported by Salt et al. (2022) Stratification of Companion Animal Life Stages from Electronic Medical Record Diagnosis Data), who, using a data-driven approach (the "Adaptive Branch Pruning Algorithm"), identified four main age clusters based on size: Youth (1–5/6 years), Midlife (6–9 years), Senior (10–13 years), and Super-senior (≥12–14 years).
Our classification therefore represents an extension of these empirically derived intervals, suitable for a data set populated largely (approximately 78%) by medium- (44%)/large- (34%) sized dogs.
- In Figure 1, correct Figure 1B where it is described "emangiosarcoma". This term is also used in other parts of the text. Please, correct
Author’s response: We corrected the typo throughout the manuscript. Anyway, we deleted figure 1 according to the other reviewers’ suggestions.
- Describe with more detail logistic regression method
Author’s response: To investigate the association between demographic and clinical variables and the probability of presenting specific splenic lesion types, we performed both univariate and multivariate logistic regression analyses. Logistic regression was selected because the outcomes of interest—presence of hemangiosarcoma (HSA) vs. all other lesions, and presence of non-tumoral (NT) lesions vs. all other lesions—were binary categorical variables.
The independent variables included age class, sex, breed type (purebred vs. mixed breed), habitat (urban, rural, other), and body size category (small, medium, large). Age and size classes were included as categorical variables using the lowest class as reference.
Adjusted odds ratios (OR) with 95% confidence intervals (CI) were calculated for all predictors. Statistical significance was set at p < 0.05. All analyses were performed using STATA 17.0 (StataCorp, College Station, TX, USA). Lines 182-191.
Results
- Survival curves show highly predictable and relatively uninformative results. For example, comparing hemangiosarcoma with non-tumor lesions provides no novel information. I recommend that survival analyses of animals with hemangiosarcoma be performed according to breed and size. Survival could also be compared between HSA and other tumors.
Author’s response: We thank the reviewer for its suggestion. Kaplan–Meier survival curves were generated to compare outcomes among dogs diagnosed with different tumor types. After excluding non-tumor lesions (NT), dogs diagnosed malignant neoplastic lesions (hemangiosarcoma (HSA), lymphoma and other sarcomas) were included in the survival analyses.
Comparison of survival curves using the log-rank test revealed significant differences across tumor categories (χ² = 10.87, p = 0.0044). Dogs with hemangiosarcoma exhibited the poorest survival probability, with curves declining more steeply than all other tumor groups.
To further quantify differences in mortality risk between tumor types, a Cox proportional hazards model was fitted using hemangiosarcoma as the reference category. The model was significant (LR χ² = 11.87, p = 0.0026) highlighting, for lymphoma, a significantly lower risk of death compared with HSA. Specifically, lymphoma was associated with a 52% reduction in hazard (HR = 0.48, 95% CI: 0.31–0.75, p = 0.001). Category “Other Sarcomas” exhibited a non-significant trend toward reduced mortality (HR = 0.80, p = 0.238).
Assessment of the proportional hazards assumption using Schoenfeld residuals indicated no significant violation (global test: χ² = 7.74, df = 4, p = 0.10), supporting the validity of the Cox model. (Line 342-361)
Subsequently, Kaplan–Meier analysis revealed no significant differences in survival among dogs with hemangiosarcoma when stratified by body size (Large, Medium, Small) and breeds (χ² = 1.20, df = 2, p = 0.55; χ² (15) = 17.42, p = 0.2942). (Line 365-368)
- The table under the survival curve shown in Figure 6 is not understood. This curve is also not well understood. Do patients with HSA have different survival rates depending on their age at diagnosis? Please, clarify
Author’s response: We thank the reviewer for this comment. Due to an error in the code used in the statistical analysis software, the survival curves and the table displayed under Figure 6 were distorted because some cases were inadvertently excluded. Despite this issue, the figure was intended to illustrate the survival of dogs diagnosed with HSA stratified by age at the time of diagnosis. However, upon further consideration, we concluded that this graph does not add meaningful information to the manuscript, as survival in dogs is influenced by multiple factors. For this reason, Figure 6 has been removed from the revised version of the manuscript.

Round 2
Reviewer 1 Report
Comments and Suggestions for Authors
I thank the authors for considering my previous comments. The detailed presentation of statistical information appears to have been useful for depicting bias in the authors' research. On the other hand, it appears that they did not consider an appropriate response to all of my previous comments.
Judging from the submitted manuscript, it is inferred that the authors place considerable emphasis on data collection methods (research materials and methods). Should this inference prove correct, this paper must be concluded to be scientifically incomplete. Firstly, the issues with the prior paper (the potential bias in data collection methods and its rationale) are not clearly presented in the manuscript. This would greatly assist in the ‘Introduction’ section, not merely listing previously reported information, but also clarifying the authors' perspective, stance, and the significance of the research to the reader. Why were specific case examples from the prior research not included in the manuscript? At least to me, the authors' intentions, as outlined in the rebuttal, do not appear sufficiently reflected within the manuscript. This is an important endeavour that must be undertaken to avoid this paper being regarded as merely a repetition of previously reported research. Provided the differentiation from previously reported research is clear, a report may be considered valuable even if the research results demonstrate no novelty whatsoever.
Furthermore, a clear methodological outline is required to address these issues. Particularly where the rationale is based on a small sample size (lines 97-102), the optimal sample size and sampling method should be explicitly presented based on statistical analysis. What is the significance of the difference between 507 and 682 cases? Should statistical justification be provided? Hypothetically, exceeding a sample size of 1,000 might enhance the credibility of the authors' claims (though this figure is based solely on my intuition and would require scientific calculation). It appears to me that they are drawing excessive conclusions from minor differences to suit their own purposes. Despite the current manuscript's emphasis on methodology, it lacks theoretical grounding. Thus, the present data rests on scientifically uncertain premises and carries a high risk of being easily refuted by future research. While the authors' analysis of a vast amount of data is commendable, but given that meaning is derived solely from the sheer volume of data, it must be regarded as a “data report” rather than a “scientific paper”.
Unfortunately, it appears that the revised manuscript submitted by the authors is not suitable for peer review. What do the different types of emphasis (blue, purple, red text and yellow background) signify?
Author Response
Reviewer: I thank the authors for considering my previous comments. The detailed presentation of statistical information appears to have been useful for depicting bias in the authors' research. On the other hand, it appears that they did not consider an appropriate response to all of my previous comments.
Judging from the submitted manuscript, it is inferred that the authors place considerable emphasis on data collection methods (research materials and methods). Should this inference prove correct, this paper must be concluded to be scientifically incomplete. Firstly, the issues with the prior paper (the potential bias in data collection methods and its rationale) are not clearly presented in the manuscript. This would greatly assist in the ‘Introduction’ section, not merely listing previously reported information, but also clarifying the authors' perspective, stance, and the significance of the research to the reader. Why were specific case examples from the prior research not included in the manuscript? At least to me, the authors' intentions, as outlined in the rebuttal, do not appear sufficiently reflected within the manuscript. This is an important endeavour that must be undertaken to avoid this paper being regarded as merely a repetition of previously reported research. Provided the differentiation from previously reported research is clear, a report may be considered valuable even if the research results demonstrate no novelty whatsoever.
Reply to reviewer:
We thank the Reviewer for this further and detailed evaluation, which raises important conceptual points regarding the framing and interpretation of our study. We acknowledge that, in the previous revision, our efforts were primarily focused on clarifying methodological transparency and statistical reporting. Considering the present comments, we recognize that the underlying scientific positioning of the manuscript required more explicit articulation within the text itself.
For this reason, the introduction has been modified to explicitly address (line 63-68; 87-91):
-the known limitations and potential biases of previous retrospective studies on canine splenic lesions (Line 63-68: “Despite this growing body of literature, reported proportions of benign versus malignant splenic lesions vary widely among studies, largely re-flecting differences in study design, referral patterns, and sample size. As a result, several assumptions commonly applied in clinical deci-sion-making remain supported by heterogeneous and sometimes un-derpowered datasets.”)
-the rationale for re-examining these issues using a substantially larger, contemporary dataset to evaluate whether conclusions derived from smaller cohorts remain stable when tested in a larger population (Line 87-91; “While the pathological spectrum of splenic lesions is well described, their relative frequencies and associated demographic risk factors remain in-consistently reported across retrospective studies, particularly those based on limited case numbers or single-center experiences.”)
Moreover, we clearly specify that: “The main goal of our study is to provide a comprehensive assessment of splenic lesions in dogs through the analysis of a sample size that is numerically superior to all the previous studies published, minimizing the influence of biases that might have influenced smaller studies and in-creasing the reliability of the observations” .
Reviewer: Furthermore, a clear methodological outline is required to address these issues. Particularly where the rationale is based on a small sample size (lines 97-102), the optimal sample size and sampling method should be explicitly presented based on statistical analysis. What is the significance of the difference between 507 and 682 cases? Should statistical justification be provided? Hypothetically, exceeding a sample size of 1,000 might enhance the credibility of the authors' claims (though this figure is based solely on my intuition and would require scientific calculation). It appears to me that they are drawing excessive conclusions from minor differences to suit their own purposes.
Reply to reviewer:
We agree that differences in sample size should not be over-interpreted. Our intention is not to claim that 682 cases represent a definitive or optimal sample size, but that increasing the sample size by approximately 35% relative to the largest previously published cohort materially reduces uncertainty in prevalence estimates and risk modeling. We avoided any language suggesting definitive conclusions.
Reviewer: Despite the current manuscript's emphasis on methodology, it lacks theoretical grounding. Thus, the present data rests on scientifically uncertain premises and carries a high risk of being easily refuted by future research. While the authors' analysis of a vast amount of data is commendable, but given that meaning is derived solely from the sheer volume of data, it must be regarded as a “data report” rather than a “scientific paper”.
Reply to reviewer:
We respectfully disagree that the manuscript constitutes a “data report” only. While the study is grounded in a large retrospective dataset, it is guided by explicit clinical and epidemiological questions, including risk stratification, outcome prediction, and reassessment of widely used diagnostic assumptions.
Reviewer: Unfortunately, it appears that the revised manuscript submitted by the authors is not suitable for peer review. What do the different types of emphasis (blue, purple, red text and yellow background) signify?
Reply to reviewer:
We apologize for this oversight. All residual highlighting and color coding, introduced during the revision process for internal tracking of changes, have now been completely removed from the manuscript.
Reviewer 3 Report
Comments and Suggestions for Authors
Thank you for the opportunity to review the revised version of the manuscript.
The revised manuscript has been submitted with tracked changes enabled, which makes careful review quite inconvenient. I would kindly ask the authors to provide a clean version without change tracking for proper evaluation.
In addition, upon a brief check, I noticed typographical errors in which “HSA” is incorrectly written as “HAS” at Lines 348, 352, 356, 400, and 426. These errors should be corrected.
Once a clean revised version is provided and these issues are addressed, I will be able to conduct a more thorough review.
Comments on the Quality of English LanguageThe English language throughout the manuscript is generally understandable, but the writing is occasionally disorganized and unpolished. Careful proofreading and editing are recommended to improve clarity, consistency, and overall readability.
Author Response
As suggested by the reviewer, we corrected the typographical error where requested. We attach the clean version of the manuscript here.
We apologise for the inconvenience.

Reviewer 4 Report
Comments and Suggestions for Authors
The authors adequately addressed the comments made. It is recommended that the manuscript be accepted for publication.
Comments on the Quality of English LanguageThe authors adequately addressed the comments made. It is recommended that the manuscript be accepted for publication.
Author Response
We thank the reviewer for his valuable suggestions which helped improve the manuscript.
Round 3
Reviewer 1 Report
Comments and Suggestions for Authors
I am grateful for your response to my previous comments. I also apologise for my failure to fully grasp the authors' research objectives. I believed that the revised manuscript clearly presents to the readers the epidemiological background and issues, as well as the authors' research objectives. One point of regret is that despite increasing the number of cases in this study, it merely reinforced findings from previously published papers, and no clear differentiation could be identified. Regrettably, within the field of veterinary medicine, and indeed across the entire modern scientific community, demonstrating novelty in research papers remains an extremely critical task, and it is a fact that the utmost effort must be devoted to this task. The authors appear to regard this report as foundational data for future research, although even adding literature review sections to this report, for example, could have provided clinicians with valuable information grounded in scientific evidence. I look forward to the authors' follow-up report.
Author Response
We thank the Reviewer for the thoughtful and constructive comments regarding the issue of novelty. We fully agree that increasing sample size alone does not constitute novelty, and we regret that this aspect was not sufficiently clarified in the previous version of the manuscript.
The aim of the present study was not merely to replicate previous findings, but to re-evaluate epidemiological patterns and risk factors for canine splenic lesions within a contemporary, histopathology-confirmed cohort using multivariate and survival analyses. We clarified this point by highlighting the gap that this manuscript aims to fill in the Introduction (line 101-121). While several results confirm earlier reports, this confirmation has clinical value considering substantial changes in diagnostic standards and population structure over recent decades. Importantly, the revised discussion now explicitly highlights novel and underexplored findings, including refined risk estimates, sex-specific differences for non-neoplastic lesions, and time-dependent survival patterns (line 478-481; 514-515; 537-544; 547-550). In addition, we investigated the literature context to provide clinicians with an evidence-based synthesis of current knowledge (line 560-580).
We hope these revisions address the Reviewer’s concerns and improve the overall quality of the manuscript.
